# Estimating Economic Losses Caused by COVID-19 under Multiple Control Measure Scenarios with a Coupled Infectious Disease—Economic Model: A Case Study in Wuhan, China

**DOI:** 10.3390/ijerph182211753

**Published:** 2021-11-09

**Authors:** Xingtian Chen, Wei Gong, Xiaoxu Wu, Wenwu Zhao

**Affiliations:** 1State Key Laboratory of Earth Surface Processes and Resource Ecology, Faculty of Geographical Science, Beijing Normal University, Beijing 100875, China; 201921051033@mail.bnu.edu.cn (X.C.); zhaoww@bnu.edu.cn (W.Z.); 2Institute of Land Surface System and Sustainable Development, Faculty of Geographical Science, Beijing Normal University, Beijing 100875, China; 3State Key Laboratory of Remote Sensing Science, College of Global Change and Earth System Science, Beijing Normal University, Beijing 100875, China

**Keywords:** COVID-19, infectious diseases model, control measures, economic losses

## Abstract

Background: The outbreak of the COVID-19 epidemic has caused an unprecedented public health crisis and drastically impacted the economy. The relationship between different control measures and economic losses becomes a research hotspot. Methods: In this study, the SEIR infectious disease model was revised and coupled with an economic model to quantify this nonlinear relationship in Wuhan. The control measures were parameterized into two factors: the effective number of daily contacts (people) (*r*); the average waiting time for quarantined patients (day) (*g*). Results: The parameter *r* has a threshold value that if r is less than 5 (people), the number of COVID-19 infected patients is very close to 0. A “central valley” around *r* = 5~6 can be observed, indicating an optimal control measure to reduce economic losses. A lower value of parameter g is beneficial to stop COVID-19 spread with a lower economic cost. Conclusion: The simulation results demonstrate that implementing strict control measures as early as possible can stop the spread of COVID-19 with a minimal economic impact. The quantitative assessment method in this study can be applied in other COVID-19 pandemic areas or countries.

## 1. Introduction

COVID-19, which shows high infectiousness and causes symptoms such as fever, cough, and shortness of breath, was first reported from Wuhan, Hubei, in mid-December 2019 [1,2,3,4]. Travel restrictions [5] and control measures [6] implemented by the Chinese government curbed the spread of the COVID-19 epidemic in Wuhan [7,8]. Stringent non-pharmacological interventions [9,10], including social distancing [11,12], masking [13], tracing, and quarantine of patients and close contacts [14,15,16], stopped the COVID-19 epidemic in China [17]. However, the control measures have inevitably caused a heavy economic burden and weakened economic vitality. Controlling the COVID-19 epidemic becomes a balancing issue, since we want to protect public health (less patients) with the least amount of economic loss [18,19].

In the first half of 2020, the GDP of the Hubei Province decreased by 19.3% year-on-year [20]. The whole year GDP in 2020 of the Hubei Province decreased by 5.0% more than in 2019 [21]. However, China’s annual GDP (gross domestic product) in 2020 increased by 2.3% over the past year, and the completion of the major objectives and tasks of socio-economic development was better than expected [22]. If the Chinese government did not implement such strict control measures, as the COVID-19 epidemic spread, we would have suffered a more serious health crisis and a heavier economic burden. However, too strict of control measures will cause unnecessary economic losses, while insufficient or inappropriate control measures will be unable to stop the COVID-19 epidemic, which threatens public health and causes larger economic losses. Consequently, it is necessary to evaluate the effectiveness and economic losses of different control measures to choose an appropriate one that can simultaneously control the epidemic and cause a low economic burden.

Firstly, the impact of the COVID-19 epidemic on the economy has been widely studied. With the spread of the COVID-19 epidemic, research on the relationship between its relevant policies as well as measures and economy will continue to be the focus of academic research [23]. For example, Brodeur et al. (2020) [23] reviewed the literature about optimal lockdown measures, influencing factors for lifting the lockdown, and the joint fiscal and monetary policy aimed at “flattening the recession curve”. Baldwin and Di Mauro (2020) [24] answered key questions about the spread speed, severity, duration of the COVID-19 epidemic, and indicated that it has the potential to derail the world economy. Unlike past natural disasters, the COVID-19 epidemic is a continuous economic shock that disrupts the activities of the labor market [25]. Moreover, the COVID-19 pandemic caused serious socio-economic effects on many aspects of the world economy, even triggering fears of recession and crisis [26]. For example, the 1918–1920 influenza pandemic resulted in 150 million deaths and triggered the Great Depression that reduced GDP by 6.0%, which provides a lesson to the economic shrinkage caused by COVID-19 [27]. The International Monetary Fund (IMF) announced in “World Economic Outlook” in April 2021 that compared with earlier forecasts, the global economic outlook had improved, but it was facing huge uncertainty [28]. Therefore, as the IMF said, the future economic trend depends on the evolution path of the health crisis [28], and it is meaningful to study the relationship between them.

Secondly, the economic losses model has been intensively studied. For example, the adaptive regional input-output (ARIO) model was used to assess the supply chain effects of different idealized lockdown scenarios, and indicated that the loss of supply chain from the COVID-19 pandemic depends on the number of countries implementing restrictions [29]. Furthermore, the loss is more sensitive to the duration of the lockdown than its strictness [29]. The multiregional input-output (MRIO) model was used to estimate the economic impacts of the California wildfire in 2018, and showed that wildfire damages totaled USD 148.5 billion, roughly 1.5% of California’s annual GDP [30]. Similarly, the input-output (IO) model is used to estimate the economic losses in Wuhan, and indicated that the monthly total economic losses (including health burden, mental health, direct losses, and indirect losses) reached CNY 177.04 billion, making up 11.06% of Wuhan’s annual GDP [31]. A cost-of-illness study estimated the economic cost of COVID-19 in 31 provinces of China, and showed that the total societal cost (mainly referred to productivity loss) of COVID-19 was CNY 2646.70 billion, and was equivalent to 2.70% of China’s annual GDP [32]. In theory, coupling economic models and infectious disease models can provide more reliable estimation results for choosing appropriate strategies [33].

The Susceptible-Exposed-Infected-Recovered infectious disease model has been intensively studied for various research objectives. For example, the early dynamics of transmission and control of COVID-19 in Wuhan were simulated with a stochastic transmission SEIR model [5]. The spread and control during the first 50 days of the COVID-19 epidemic in China was investigated with the SEIR model [6]. A reconstruction of the full transmission dynamics of COVID-19 in Wuhan was modelled with an extended SEIR model [7]. Moreover, the effect of mobility restrictions on the spread of COVID-19 in Shenzhen was quantified with an SEIR model that aggregated mobile phone data [12]. The risk assessment of COVID-19 in China was estimated by combining urban spatiotemporal big data and the SEIR model [34]. The effect of anti-contagion policies on the COVID-19 pandemic in six countries was assessed with a simple SIR model [35]. On the other hand, some studies have tried to associate the economic loss (such as cost and burden) with control measures (such as lockdown and social distancing) by the infectious disease model. For example, an SIR model was built to quantify the tradeoff between severity and timing, and showed that when the infections in the total population exceed 1%, the economic burden (staffing shortages) of the epidemic over an 18-month period will reach 10% [36]. Alvarez et al. (2020) [37] used the SIR model to study the optimal lockdown policy, and showed that the absence of testing would increase the economic cost of the lockdown and delay the duration of the lockdown. A multi-group SIR model was built to study targeted lockdown, and suggested that reducing social interactions between different groups and increasing the screening and isolation can minimize economic losses and deaths [38].

In summary, there are many studies about COVID-19 control measures. For example, the full transmission dynamic of COVID-19 in Wuhan across five periods [8] was reconstructed with an extended SEIR model [7]. Multiple studies have estimated economic losses caused by the COVID-19 epidemic. For example, the economic cost of COVID-19 in China was estimated by health-care costs (diagnosis and treatment), non-health-care costs (compulsory quarantine), and productivity losses (lost work time) [32]. However, few studies estimate the economic losses caused by the COVID-19 epidemic under different control measures. Therefore, how to quantify the relationship between COVID-19 control measures and economic losses caused by the COVID-19 epidemic is still a key research question. This study aims to develop a model to parameterize the control measures and quantify economic losses caused by both the COVID-19 epidemic and the adopted control measures. 

The paper is organized as follows: Section 1 introduces the literature review about economic losses caused by the COVID-19 epidemic, the research gap, and the research objective of this study. Section 2 introduces the analysis methodology of this study, including the infectious disease model (see Section 2.1), the economic losses model (see Section 2.2), and their parameters (see Table 1, Table 2 and Table 3). Section 3 introduces the infectious disease model verification (see Section 3.1), and the influence of social distancing (*r*) and tracking/quarantining ability (*g*) on public health (see Section 3.2) and economic losses (see Section 3.3). Section 4 compares our findings with previous studies, then summarizes the improvements and limitations of this study (see Section 4.1), and also proposes some suggestions on how to effectively control COVID-19 spread with less economic losses (see Section 4.2). Section 5 introduces the conclusion of this study.

## 2. Materials and Methods

This study focuses on parameterizing the control measures into two key factors, and then quantitatively evaluates economic losses caused by both the COVID-19 epidemic and the adopted control measures. The analysis of this study mainly includes the following steps. First, the infectious disease model was revised to parameterize control measures into two factors (see Section 2.1.1): (1) the level of social distancing, represented by the effective number of daily contacts (parameter *r*); (2) the level of tracing and quarantining the infected patients and their close contacts, represented by the average waiting time for quarantined patients (parameter *g*). Second, the revised model was used to simulate the spread process of the COVID-19 epidemic in Wuhan to further validate its accuracy (see Section 2.1.2). Third, we quantitatively analyzed the effect of COVID-19 on both human health and economic losses. The estimated economic losses of COVID-19 include the cost of curing infected patients (see Section 2.2.1), the expenditure to quarantine patients and close contacts (see Section 2.2.2), inoperable losses caused by patients and quarantined people that are unable to work (see Section 2.2.3), and social distancing losses caused by reducing the activity of social interactions (see Section 2.2.4). Finally, we quantified economic losses under various control measures (see Section 2.2.5).

### 2.1. The Infectious Disease Model

#### 2.1.1. The Framework of the Infectious Disease Model

Firstly, Figure 1 explains the framework of the infectious disease model in this study. The classical Susceptible-Exposed-Infected-Recovered (SEIR) model can simulate the epidemic’s dynamic transmission mechanism with differential equations and describes the flow between four main compartments: susceptible (S), exposed (E), infected (I), and recovered (R) [39]. Based on the work of Hao et al. (2020) [7], which extended the classical SEIR model with three additional compartments: presymptomatic (P), asymptomatic (A), and hospitalized (H), we developed the model by adding a compartment of quarantined (Q) (see Figure 1). 

Moreover, considering the degree of self-protection (including social distancing, masking, and disinfecting), we parameterized susceptible peoples’ risk of being infected. On the other hand, considering the effectiveness of quarantine (including medical resources and emergency response), we parameterized the capacity of isolating infected patients. Therefore, the transmission dynamics of COVID-19 can be described with the first-order differential equations, as shown in Equation (1).

(1)$$\left\{ \begin{array}{l} \frac{dS}{dt}=n-r\left(1-\theta \right)\beta \left(\alpha P+\alpha A+I\right)\frac{S}{N}-n\left(\frac{S}{S+E+P+A+R}\right)\\ \frac{dE}{dt}=r\left(1-\theta \right)\beta \left(\alpha P+\alpha A+I\right)\frac{S}{N}-\frac{1}{D_{e}}E-n\left(\frac{E}{S+E+P+A+R}\right)\\ \frac{dP}{dt}=\frac{1}{D_{e}}E-\frac{1}{D_{p}}P-n\left(\frac{P}{S+E+P+A+R}\right)\\ \frac{dA}{dt}=\left(1-p\right)\frac{1}{D_{p}}P-\frac{1}{D_{i}}A-n\left(\frac{A}{S+E+P+A+R}\right)\\ \frac{dI}{dt}=p\frac{1}{D_{p}}P-\frac{1}{D_{i}}I-gI\\ \frac{dQ}{dt}=gI-\frac{1}{D_{i}}Q-\frac{1}{D_{i}}Q\\ \frac{dH}{dt}=\frac{1}{D_{i}}Q-\frac{1}{D_{h}}H\\ \frac{dR}{dt}=\frac{1}{D_{i}}\left(I+A+Q\right)+\frac{1}{D_{h}}H-n\left(\frac{R}{S+E+P+A+R}\right) \end{array}\right.$$
where r is the effective number of daily contacts; β is the SARS-CoV-2 infection rate; θ is the degree of self-protection; p is the ascertainment rate; α is the ratio of spread rate for unconfirmed over confirmed cases; $D_{p}$ is the presymptomatic infectious period; $D_{e}$ is the latent period; $D_{i}$ is the symptomatic infectious period; g is the average waiting time for quarantined patients; $D_{h}$ is the average hospitalization period; N is the total population in Wuhan; n is the daily domestic inbound and outbound travelers in Wuhan.

#### 2.1.2. The Five-Stage Model and its Parameters

Secondly, the first-order differential equations (Equation (1)) were divided into five stages. In the following full text, we prefer to call the infectious disease model in this study “the five-stage model” or “the developed model”. Based on an earlier study of Pan et al. (2020) [8], the early period of the COVID-19 outbreak in Wuhan was divided into five stages: 1–9 January (before Chinese Spring Festival Transportation), 10–22 January (Chinese Spring Festival Transportation), 23 January–1 February (during lockdown of Wuhan), 2–16 February (centralized isolation and quarantine), and 17 February–8 March (community screening and testing).

Next, the parameter of the five-stage model was introduced in detail. On one hand, most of the parameter’s values of the five stages referred to the study by Hao et al. (2020) [7] and other original studies, which are listed in Table 1. For example, the total population in Wuhan $\left(N\right)$ is set to be 10,000,000 people [7,40]. The daily domestic inbound and outbound travelers in Wuhan $\left(n\right)$ in the five stages is set to be 500,000; 800,000; 0; 0; 0 people, respectively [7,40]. The ascertainment rate (p) in the five stages is estimated to be 0.15, 0.15, 0.14, 0.10, and 0.16, respectively [7]. The ratio of the spread rate for unconfirmed (including patients with pre-symptoms) over confirmed cases $\left(\alpha \right)$ is estimated to be 0.55 [41], which is lower than the infection rate of confirmed cases. The incubation period of confirmed cases is 5.2 days [3]. The transmission rate during the presymptomatic stage is 44% [42]. Consequently, the presymptomatic infectious period $\left(D_{p}\right)$ is 2.3 (5.2 × 0.44 = 2.3) days [7,42], and the latent period $\left(D_{e}\right)$ is 2.9 (5.2 − 2.3 = 2.9) days [3,7]. Assuming that the infectivity of presymptomatic cases is consistent with the infectivity of symptomatic cases [14], the symptomatic infectious period $\left(D_{i}\right)$ is also 2.9 days [7,14], and the average hospitalization period $\left(D_{h}\right)$ is 10 days [4].

The effective number of daily contacts $\left(r\right)$ and the average waiting time for quarantined patients (g) in the five stages were estimated. The transmission rate of confirmed cases $\left(b\right)$ [7] was mainly depended on three key variables (b=r×β×θ): the effective number of daily contacts $\left(r\right)$, the SARS-CoV-2 infection rate $\left(\beta \right)$, and the degree of self-protection $\left(\theta \right)$. The SARS-CoV-2 infection rate $\left(\beta \right)$ was assumed to be stable in the early COVID-19 outbreak, and was 0.395 in the five stages [5]. The degree of self-protection $\left(\theta \right)$ among susceptible people was assumed to be time-varying in the five stages: 0.6, 0.6, 0.7, 0.8, and 0.9, respectively. 

Therefore, the effective number of daily contacts $\left(r=\frac{b}{\beta \theta }\right)$ in the five stages was estimated to be 8.29, 8.29, 3.37, 2.15, and 2.53 people, respectively. Then, by fitting with COVID-19 confirmed cases in Wuhan, the average waiting time for quarantined patients (g) was estimated to be 3.5, 1.7, 2.0, 1.7, and 1.2 days, respectively. 

### 2.2. The Economic Losses Model

The economic losses caused by COVID-19 can be categorized into four parts: (1) treatment costs of infected patients (see Section 2.2.1), which are proportional to the number of infected patients; (2) quarantine-related expenditures (see Section 2.2.2), which are proportional to the number of quarantined patients and their close contacts; (3) inoperable losses (see Section 2.2.3), which are due to the absence of work (patients at hospitals, patients staying at home but unable to work, and patients and close contacts quarantined); (4) social distancing losses (see Section 2.2.4), which are due to reducing the activity of social interactions (causing the supply shock and the demand shock to the industry sectors). 

Among the above four parts of economic losses, the first three can be assumed to have a linear relationship with the number of patients [32]. Social distancing measures caused the reduction in contact activity that related to the fall in labor supply and consumer demand, which led to the decrease in GDP [24,28]. The supply shock and demand shock are different for different sectors [24]. Thus, the last one, social distancing losses, was assumed to have a non-linear relationship with the level of contact activity [32,43] (represented by the effective number of daily contacts).

#### 2.2.1. Treatment Costs

Firstly, treatment costs were mainly related to the number of infected patients. We simulated the cumulative number of infected patients under various control measures $\left(I_{i}\right)$. According to the different clinical severity of COVID-19 hospitalized patients [4] and their corresponding treatment costs [44], we respectively estimated the treatment costs of infected patients with no complications $\left(C_{1}\left(1-P_{1}\right)I_{i}\right)$, with complications $\left(C_{2}P_{1}\left(1-P_{2}\right)I_{i}\right)$, and with severe complications $\left(C_{3}P_{1}P_{2}I_{i}\right)$. Hence, the total treatment costs of infected patients under various control measures (r and g) were estimated with Equation (2). The parameters are listed in Table 2.
(2)$$Loss_{treatment}=C_{1}\left(1-P_{1}\right)I_{i}+C_{2}P_{1}\left(1-P_{2}\right)I_{i}+C_{3}P_{1}P_{2}I_{i}$$
where $C_{1}\left(1-P_{1}\right)I_{i}$ is the treatment cost of COVID-19 infected patients with no complications; $C_{1}$ is the average treatment cost of a COVID-19 infected patient with no complications; $P_{1}$ is the probability of a COVID-19 infected patient being admitted to the Intensive Care Unit (ICU); $I_{i}$ is the cumulative number of infected patients under various control measures (r and g); $C_{2}P_{1}\left(1-P_{2}\right)I_{i}$ is the treatment cost of COVID-19 infected patients with complications; $C_{2}$ is the average treatment cost of a COVID-19 infected patient with complications; $P_{2}$ is the probability of Acute Respiratory Distress Syndrome (ARDS) after a COVID-19 infected patient is admitted to the Intensive Care Unit (ICU); $C_{3}P_{1}P_{2}I_{i}$ is the treatment cost of COVID-19 infected patients with severe complications; $C_{3}$ is the average treatment cost of a COVID-19 infected patient with severe complications.

#### 2.2.2. Quarantine-Related Expenditures

Secondly, quarantine-related expenditures were mainly related to the number of quarantined patients. We simulated the cumulative number of quarantined patients during the COVID-19 outbreak in Wuhan $\left(Q_{basic}\right)$ with Equation (1). We assumed that expenditures for disaster prevention and emergency management in the Hubei province in the first quarter of 2020 $\left(S\right)$ were mainly spent in Wuhan. Since the number of close contacts can be estimated by multiplying the effective number of daily contacts, the number of days before quarantine, and the number of quarantined patients, the total quarantine-related expenditures were assumed to be proportional to the cumulative number of quarantined patients $\left(Q_{i}\right)$. Hence, quarantine-related expenditures under various control measures (r and g) were estimated with Equation (3). The parameters are listed in Table 2.
(3)$$Loss_{quarantine}=S\frac{Q_{i}}{Q_{basic}}$$
where S is the total expenditures for disaster prevention and emergency management in the Hubei province in the first quarter of 2020, which is assumed to be approximately proportional to the number of quarantined patients; $Q_{basic}$ is the cumulative number of quarantined patients during the COVID-19 outbreak in Wuhan; $Q_{i}$ is the cumulative number of quarantined patients under various control measures (r and g).

#### 2.2.3. Inoperable Losses

Thirdly, inoperable losses were mainly estimated by two parts. One part was that infected patients who were unable to work due to receiving treatment in the hospital (the time from first symptom to ARDS was eight days [4] and the hospital stay was ten days [4]) and self-isolation at home (14 days) $\left(GDP_{per}T_{I}I_{i}\right)$. The other part was that quarantined patients who were unable to work due to receiving quarantine (14 days) and self-isolation at home (14 days) $\left(GDP_{per}T_{Q}Q_{i}\right)$. Hence, the total inoperable losses under various control measures (r and g) were estimated with Equation (4). The parameters are listed in Table 2.
(4)$$Loss_{inoperable}=GDP_{per}T_{I}I_{i}+GDP_{per}T_{Q}Q_{i}$$
where $GDP_{per}T_{I}I_{i}$ is the infected patients who were unable to work due to receiving treatment in the hospital and self-isolation at home; $GDP_{per}$ is the per capita daily GDP in 2019; $T_{I}$ is the average time that infected patients received treatment (8 + 10 = 18 days) and self-isolation (14 days); $GDP_{per}T_{Q}Q_{i}$ is the quarantined patients who were unable to work due to receiving quarantine and self-isolation at home; $T_{Q}$ is the average time that quarantined patients received quarantine (14 days) and self-isolation (14 days).

#### 2.2.4. Social Distancing Losses

Lastly, social distancing losses are caused by reducing the activity of social interactions (represented by the effective number of daily contacts (r)). Unlike other sources of losses, social distancing losses have a strong nonlinear relationship with the level of contact activity [32,43], which is represented by the effective number of daily contacts (r). The parameters are listed in Table 3. 

The effective number of daily contacts before the COVID-19 outbreak $\left(r_{0}\right)$ was estimated by the contact’s number matrix with different age structures in China $\left(C_{i}\right)$ [47] and the proportion of the age structure in China in 2019 $\left(P_{i}\right)$ [46], which is shown in Equation (5).
(5)$$r_{0}=\overset{16}{\underset{i}{\sum }}C_{i}P_{i}$$
where $r_{0}$ is the effective number of daily contacts before the COVID-19 outbreak; $C_{i}$ is the contact’s number matrix with intervals of five years of age in China; $P_{i}$ is the proportion of the age structure with intervals of five years of age in China in 2019; i is the index of the five years of age interval.

The weighted value of the effective number of daily contacts during the COVID-19 outbreak $\left(\bar{r}\right)$ was then estimated by the effective number of daily contacts in each stage of the COVID-19 epidemic in Wuhan $\left(r_{i}\right)$ and the proportion of the duration in each stage of the COVID-19 epidemic in Wuhan $\left(D_{i}\right)$ [7], which is shown in Equation (6).
(6)$$\bar{r}=\overset{5}{\underset{i}{\sum }}r_{i}D_{i}$$
where $\bar{r}$ is the weighted value of the effective number of daily contacts during the COVID-19 outbreak; $r_{i}$ is the effective number of daily contacts in each stage of the COVID-19 epidemic in Wuhan (see Table 1), which is 8.29, 8.29, 3.37, 2.15, and 2.53 people, respectively; $D_{i}$ is the proportion of the duration in each stage of the COVID-19 epidemic in Wuhan (see Table 1), which is 13.24%, 19.12%, 14.71%, 22.06%, and 30.88%, respectively; i is the index of stage of the COVID-19 epidemic in Wuhan.

Moreover, the basis of the GDP value, which is assumed to be the expected seasonal GDP if COVID-19 did not break out, is estimated with the Autoregressive Integrated Moving Average model (ARIMA) [48]. To facilitate future calculation, only the GDP growth rate is considered. 

Social distancing losses caused by reducing the effective number of daily contacts from a normal level are assumed to be proportional to ${\left(\frac{\bar{r}}{r_{0}}\right)}^{\alpha }$, where $\frac{\bar{r}}{r_{0}}$ is the reduction in contact activity (31.73%) caused by the COVID-19 outbreak in Wuhan, and α is the degree of nonlinearity in each industry sector’s response to the reduction in contact activity $\left(\frac{\bar{r}}{r_{0}}\right)$.

The value of the nonlinearity index α of each industry sector (see Figure 5) can be estimated with Equations (7) and (8), where $C_{2019}$ is the cumulative output value of each industry sector in the Hubei province in the first quarter of 2020 [45]; $GDP_{p}$ is the actual growth rate of each industry sector in the Hubei province in the first quarter of 2020 [45]; $\tilde{GDP_{p}}$ is the expected growth rate of the industry sector in the Hubei province in the first quarter of 2020, which was simulated with the ARIMA model [48].
(7)$$C_{2019}\left(1+GDP_{p}\right)=C_{2019}\left(1+\tilde{GDP_{p}}\right){\left(\frac{\bar{r}}{r_{0}}\right)}^{\alpha }$$
(8)$$\alpha =\log_{\left(\frac{\bar{r}}{r_{0}}\right)}\left(\frac{1+GDP_{p}}{1+\tilde{GDP_{p}}}\right)$$

Given each industry sector’s nonlinearity index to the COVID-19 epidemic in Wuhan (Equation (7)), the expected growth rate of each industry sector under the different level of contact activity ($GDP_{p}{}^{r}$) was estimated by Equation (9). The social distancing losses of each industry sector are assumed to be equal to the difference between the actual growth rate and the expected growth rate under the different level of contact activity $\left(\frac{r}{r_{0}}\right)$. Therefore, the social distancing losses of each industry sector in Wuhan that suffered from the COVID-19 epidemic in a different parameter r (the effective number of daily contacts) can be estimated with Equation (10).
(9)$$GDP_{p}{}^{r}=\left(1+\tilde{GDP_{p}}\right){\left(\frac{r}{r_{0}}\right)}^{\alpha }-1$$
(10)$$Loss_{distancing}=PC_{2019}\left(GDP_{p}{}^{r}-\tilde{GDP_{p}}\right)$$
where r is the effective number of daily contacts, ranging from 0 to $r_{0}$ people; $\frac{r}{r_{0}}$ is the different level of contact activity; $GDP_{p}{}^{r}$ is the expected growth rate of each industry sector under the different level of contact activity; P is the percentage of the output value in 2019 Wuhan to the output value in the 2019 Hubei province [45]; $\left(GDP_{p}{}^{r}-\tilde{GDP_{p}}\right)$ is the total losses rate of each industry sector that suffered from the COVID-19 impact with the different level of contact activity.

#### 2.2.5. Total Economic Losses

Figure 2 introduces the framework for estimating the total economic losses under various control measures. The treatment costs ($Loss_{treatment}$), quarantine-related expenditures ($Loss_{quarantine}$), and inoperable losses ($Loss_{inoperable}$) were estimated under various control measures (r and g). The social distancing losses ($Loss_{distancing}$) under the different levels of contact activity (r) were estimated. Consequently, the total economic losses ($Loss\left(r,g\right)$) under various control measures (r and g) were estimated with Equation (11).
(11)$$Loss\left(r,g\right)=Loss_{treatment}+Loss_{quarantine}+Loss_{inoperable}+Loss_{distancing}$$

### 2.3. Data

The parameters of the infectious disease model in five stages are presented in Table 1. The parameters of estimating economic losses are presented in Table 2 and Table 3. Moreover, the COVID-19 confirmed cases data in Wuhan comes from The Chinese Centers for Disease Control and Prevention (Available online: http://2019ncov.chinacdc.cn/2019-nCoV/) (accessed on 16 September 2021). The economy-related monthly data of each industry sector is provided by the Hubei Provincial Statistics Bureau (Available online: http://tjj.hubei.gov.cn/tjsj/sjkscx/tjyb/tjyb2021/) (accessed on 16 September 2021).

## 3. Results

### 3.1. The Infectious Disease Model Verification

With the calibrated parameters in the five stages (Table 1), simulated cases from the five-stage model were verified with the COVID-19 confirmed cases in Wuhan (Figure 3). The simulated cases can fit the reported cases sufficiently, except in the third stage (from 23 January to 2 February), where a significant deviation can be observed. There are two factors causing this. The first is the peak number of reported cases (the peak point in Figure 3). The COVID-19 epidemic peaked in the third stage, while the five-stage model did not capture the peak point and kept the trend upwards. The second is the medical resources. At this stage, due to limited medical resources, the capacity of hospitalization and quarantine was weakened.

The simulated results showed that as the effective number of daily contacts $\left(r\right)$ increased, the COVID-19 epidemic in Wuhan became hard to be contained. In the mid-term of the COVID-19 spread, when the parameter r≈14, the daily increase in the number of infected patients accounts for about 1.1% of the total population (see Appendix A), and the daily increase in the number of quarantined patients accounts for about 0.8% of the total population (see Appendix A). Moreover, with the spread of COVID-19, the number of confirmed cases exceeded the affordability of local medical resources, which forced the government to carry out more stringent control measures to slow down the spread of COVID-19. The degree of strictness of control measures can be parameterized into two factors: the effective number of daily contacts $\left(r\right)$, which represents the degree of social distancing, and the average waiting time for quarantined patients $\left(g\right)$, which represents the ability of tracing and quarantining infected patients. Therefore, the effectiveness of multiple control measures represented by various combinations of gridded r and g values can be evaluated comprehensively, as addressed in the following sections.

### 3.2. The Influence of Social Distancing (r) and Tracking/Quarantining Ability (g) on Public Health

The cumulative number of infected patients was calculated based on multiple scenarios of the parameter r and g, and its contour map is shown in Figure 4. All other parameters of the five-stage model were set to the values of stage-2 (see Table 1). As shown in Figure 4, the cumulative number of COVID-19 infected patients increases with the parameter r, but is almost parallel to x-axis if g is large. Clearly, to reduce the cumulative number of infected patients, simultaneously reducing r and g is preferred. The effectiveness of r on COVID-19 spreading is more significant, especially when g is large; if g is small, both r and g are effective in stopping the spread of COVID-19. The parameter r has a threshold value that, if r is less than 5 (people), the number of COVID-19 infected patients is very close to 0.

### 3.3. The Influence of Social Distancing (r) and Tracking/Quarantining Ability (g) on Economic Losses

Firstly, treatment costs (see Appendix A), quarantine-related expenditures (see Appendix A), and inoperable losses (see Appendix A) under various control measures (r and g) were estimated with Equations 2, 3, and 4, respectively. We confirmed that the treatment costs were proportional to the number of infected patients, and also that the quarantine-related expenditures were proportional to the number of quarantined patients. 

Based on each industry sector’s response pattern to the COVID-19 impact (see Section 2.2.4 for details), we estimated each industry sector’s response to different contact activity levels, represented by the parameter r. The industry sector’s response patterns to the COVID-19 impact can be divided into three types (Figure 5): (1) The nonlinearity index α is less than 1. The level of contact activity represented by the parameter r has a slight effect on some industry sectors, because these sectors do not strongly rely on a lot of manpower or face-to-face communication. For example, the index α of chemical products is only 0.674, because operating modern automatic chemical equipment only requires a few people. (2) The nonlinearity index α approximately equals to 1. In this case, the value of parameter r linearly affects some industries, such as the non-metallic mineral manufacturing industry sector, the equipment manufacturing industry sector, and the public management industry sector. (3) The nonlinearity index α is larger than 1. In this case, the value of parameter r significantly affects the productivity of some industries, such as agriculture, mining, construction, and most tertiary industries. For example, the α of building and finance is 4.387 and 2.841, respectively. 

Finally, the total economic losses in Wuhan under various gridded control measures (r and g) were estimated (Figure 6). Generally, economic losses are sensitive to parameter r. A “central valley” around r=5~6 can be observed, indicating that there is an optimal region to reduce economic losses. If r is large, the treatment costs and inoperable losses become very large; if r is small, the social distancing losses are large. If the parameter g is large, the contour lines are nearly parallel to the x-axis, indicating that reducing g is effective if g is small enough, but useless if g is large. Therefore, it is necessary to track and quarantine infected patients on time.

## 4. Discussion

### 4.1. Literature-Based Verification

Firstly, the parameter r has a significant impact on the time required to reach the peak of the COVID-19 infection curves (see Appendix A). It is believed that restrictions on social activities have delayed the outbreak of COVID-19 [49,50]. On the other hand, the parameter g can influence the scale of the COVID-19 infection curves (see Figure 4). A larger parameter g, which means a lower capacity for tracing and quarantining potential patients, leads to the more serious spread of COVID-19. This result verifies that high-intensity isolation measures are very important to stopping the spread of COVID-19 [51]. 

Secondly, the cumulative number of infected patients and quarantined patients can be used as indicators of the effectiveness of control measures against the COVID-19 spread. As shown in Figure 1, the number of infected patients increases towards the direction of larger parameters r and g (lax control measures), and its growth rate is accelerating. This finding confirms that, in the early stage of the COVID-19 spread, a lacking in non-pharmaceutical interventions facilitates the spread of COVID-19 [52]. When the parameter g is small, as the parameter r increases, there is still a risk of the COVID-19 outbreak (see Figure 4). This finding confirms that, when implementing non-pharmaceutical interventions with strict control capabilities, relaxing the restriction on social activities increases the risk of COVID-19 recurrence [53,54]. Moreover, when the parameter r is less than about six people, the COVID-19 outbreak is stopped, indicating that non-pharmaceutical interventions are effective in stopping the spread of COVID-19 (see Figure 4). Therefore, if unknown infectious factors or ways increase the transmission rate of COVID-19 [55], it is necessary to carry out stringent control measures to control the epidemic. This conclusion agrees well with a previous study [7].

Thirdly, some industry sectors’ response to the reduction in contact activity is insensitive (see Figure 5), such as working online or manufacturing that have little social interactions, which provide vitality for economic recovery. This finding confirms that with the popularity of social distancing and remote working, the digital economy can help mitigate the economic losses from COVID-19, for example, in Kenya [56]. Besides, by emphasizing the need for cooperation with the government, civil society, and private individuals, the case study in Vietnam offers valuable lessons for responding to a public health crisis [57].

Finally, a “central valley” around r = 5~6 can be observed, indicating that there is an optimal region to reduce economic losses (see Figure 6). Among the economic losses estimated with Equation (11) in our study, the minimum value was CNY 188.81 billion, which was equivalent to 11.63% of Wuhan’s GDP and to 0.19% of China’s GDP. You et al. [31] estimated that monthly total economic losses (including health burden, mental health, direct losses, and indirect losses) reached CNY 177.04 billion, and made up 11.06% of Wuhan’s GDP. Moreover, Jin et al. [32] estimated that the total societal cost (mainly referred to productivity loss) of COVID-19 in China was CNY 2646.70 billion, and was equivalent to 2.70% of China’s GDP. The total cost of the pandemic is estimated to be approximately 90% of the annual GDP of the United States [58]. Moreover, the impact of factors (such as demography, contact patterns, disease severity, and health care capacity) on the COVID-19 suppression strategies in low- and middle-income countries [59], and the sensitivity of patients (such as demographic and socioeconomic-cognitive) to health costs [60], both need to be considered.

### 4.2. Suggestion on Control of COVID-19 Spread

On one hand, the stringent control measures help to reduce costs and losses caused by the COVID-19 epidemic. On the other hand, the early establishment of an emergency response strategy helps to mitigate the economic burden caused by COVID-19. Consequently, to control the COVID-19 outbreak and mitigate the economic impact caused by the COVID-19 epidemic, we should develop a prompt and strict emergency strategy to stop the spread of COVID-19. This could simultaneously realize stopping the COVID-19 spread, minimizing the economic shock, and providing vitality for economic recovery.

### 4.3. Contribution and Limitation

This study presents three novel contributions, as follows. First, a novel infectious disease model has been developed based on coupling the infectious disease model and the economic model to quantify economic losses caused by COVID-19 in Wuhan. Second, the control measures were parameterized into two important parameters: the effective number of daily contacts (r), and the average waiting time for quarantined patients (g). Third, a new method for analyzing the nonlinear response relationship of the industry sector to the impact of COVID-19 and the degree of control measures was proposed. The responses of total infected patients and total economic losses to the two parameters were simulated and visualized as contour maps (see Figure 6). The quantitative assessment method in this study can apply to other areas/countries that are still fighting COVID-19, or to related studies on other infectious diseases.

However, some limitations should be noted in this study. First, the imbalance between the domestic inbound and outbound population size in Wuhan is not considered in this study, which is a further-up work for us in the future. Second, the close contacts of quarantined patients were also isolated, but we do not have the accurate data. So, we did not consider this in our method. Third, the applicability of the five-stage model to a larger spatial scale needs validation with reported cases in larger areas. Lastly, the impact of factors, such as socio-economic conditions and healthcare conditions, should be studied in future work.

## 5. Conclusions

This study developed a new infectious disease model by coupling the infectious diseases model and the economic losses model. Based on the developed model, we quantitatively evaluated the total economic losses under various control measures for combatting COVID-19. The main conclusions are as follows: (1) strict social distancing (less than five daily contacts, represented by parameter r) can drastically reduce the total economic losses caused by the COVID-19 epidemic, and can stop the COVID-19 spread in a short time. Although it may cost short-term economic recession, social distancing is indeed effective to protect public health. (2) The “optimal” control measure lies in the “central valley”, where the effective number of daily contacts (r) is 5~6 people, and the average waiting time for quarantined patients (g) should be as short as possible. With the increasing level of emergency responses (the lower parameter g), the minimum value of total economic losses becomes smaller; the parameter r that can minimize the total economic losses increases with the lower parameter g. Therefore, we suggest implementing strict control measures as early as possible. This not only maintains economic vitality for recovery, but also minimizes the economic burden.

## Figures and Tables

**Figure 1 ijerph-18-11753-f001:**
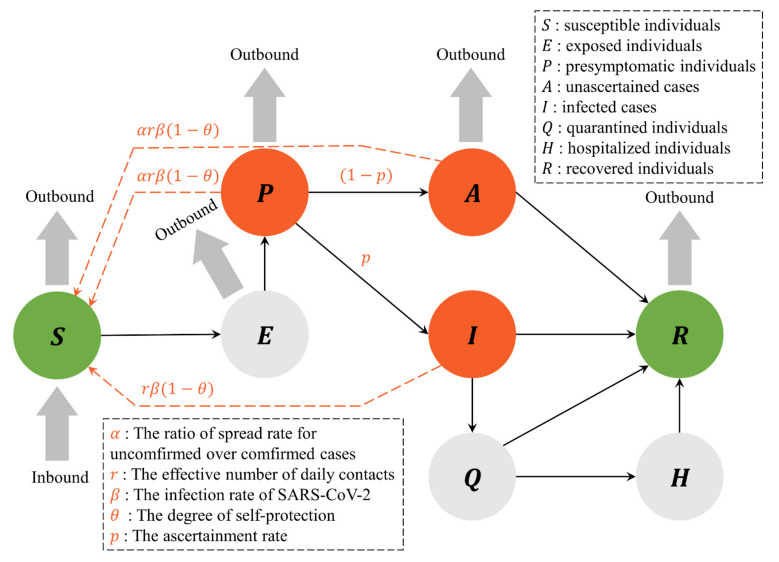
The framework of the infectious disease model. The black solid line indicates the transmission dynamics of the COVID-19 epidemic in each compartment.

**Figure 2 ijerph-18-11753-f002:**
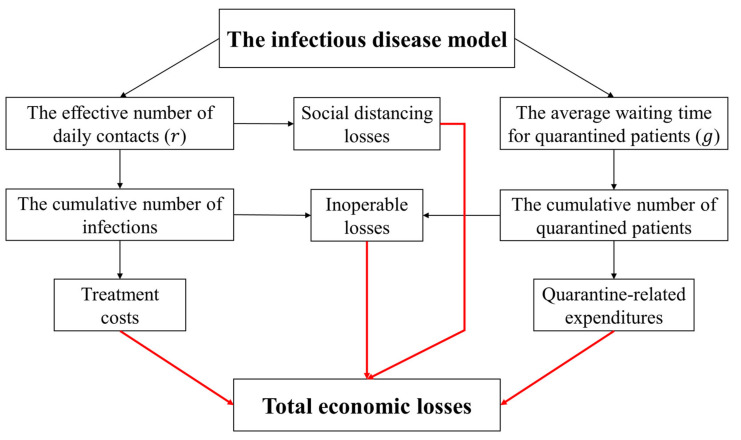
The framework of estimating total economic losses under various control measures.

**Figure 3 ijerph-18-11753-f003:**
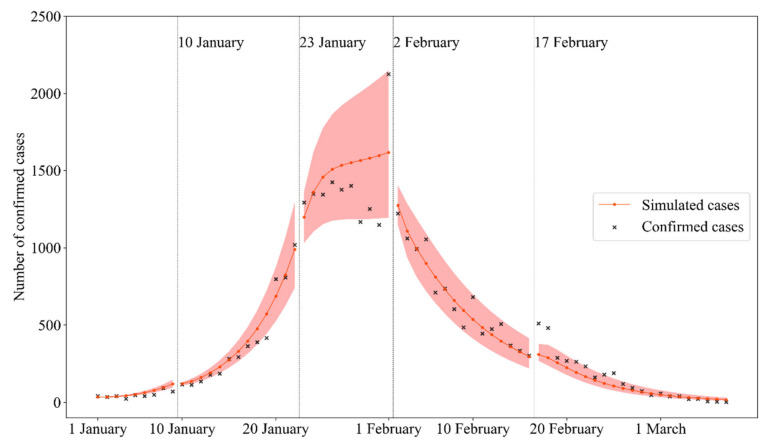
Simulated cases from the five-stage model were verified with the COVID-19 confirmed cases in Wuhan. The confirmed case data comes from The Chinese Centers for Disease Control and Prevention. The fitting technique of the five-stage model was introduced in Section 2.1.2.

**Figure 4 ijerph-18-11753-f004:**
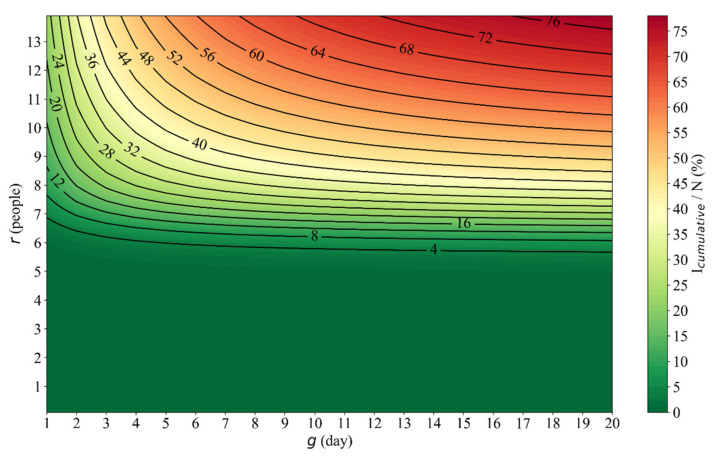
The contour map of the cumulative number of COVID-19 infected patients in Wuhan, where r is the effective number of daily contacts; g is the average waiting time for quarantined patients; $I_{cumulative}$ is the cumulative number of infected patients; N is the total population.

**Figure 5 ijerph-18-11753-f005:**
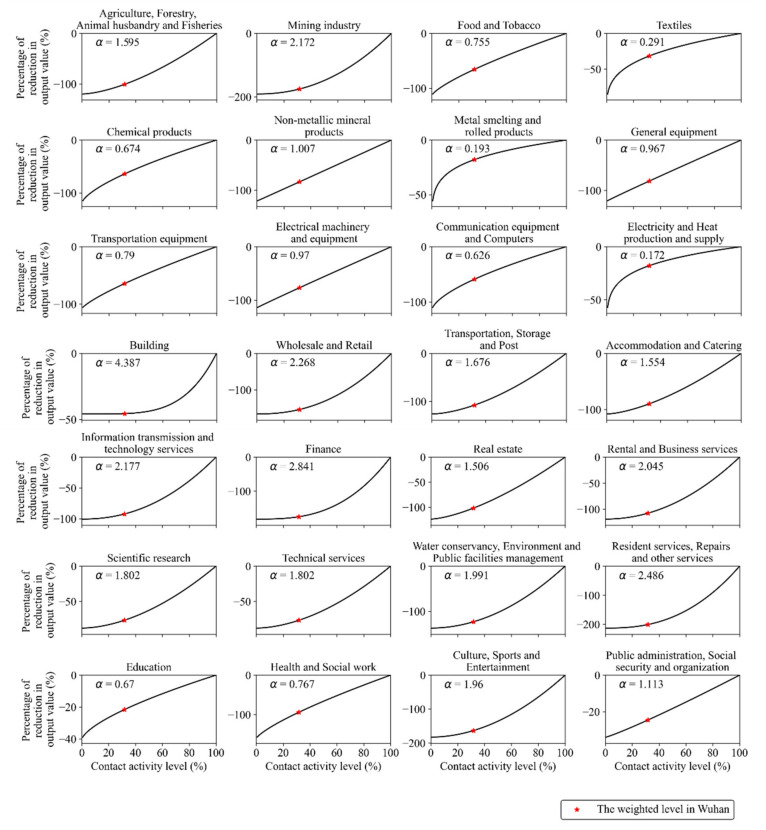
Reduction percentage of each industry sector under the different level of contact activity in Wuhan. The weighted level in Wuhan is the reduction in contact activity caused by the COVID-19 outbreak in Wuhan $\left(\frac{\bar{r}}{r_{0}}\right)$, is 31.73% (see Equations (5) and (6)); α is the degree of nonlinearity of each industry sector’s response to the reduction in contact activity $\left(\frac{\bar{r}}{r_{0}}\right)$ (see Equation (8)). When the contact activity level reaches 100% (r = $r_{0}$ ≈ 13.96), it means returning to the level before the COVID-19 outbreak in Wuhan.

**Figure 6 ijerph-18-11753-f006:**
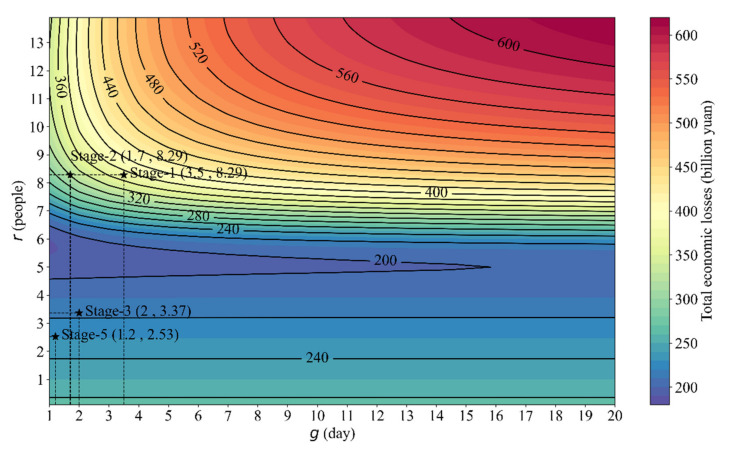
The contour map of total economic losses in Wuhan under various gridded control measures (r and g), where r is the effective number of daily contacts; g is the average waiting time for quarantined patients. Total economic losses ranged from CNY 188.81 to 618.70 billion.

**Table 1 ijerph-18-11753-t001:** Parameters of the infectious disease model in five stages.

	Meaning	1.1–1.9	1.10–1.22	1.23–2.1	2.2–2.16	2.17–3.8	Reference
b	The transmission rate of confirmed cases	1.31	1.31	0.40	0.17	0.10	[7]
r	The effective number of daily contacts (people)	8.29	8.29	3.37	2.15	2.53	Estimated
β	The SARS-CoV-2 infection rate	0.395	0.395	0.395	0.395	0.395	[5]
θ	The degree of self-protection	0.6	0.6	0.7	0.8	0.9	Assumed
p	The ascertainment rate	0.15	0.15	0.14	0.10	0.16	[7]
α	The ratio of spread rate for unconfirmed over confirmed cases	0.55	0.55	0.55	0.55	0.55	[41]
$D_{p}$	The presymptomatic infectious period	2.3	2.3	2.3	2.3	2.3	[7,42]
$D_{e}$	The latent period	2.9	2.9	2.9	2.9	2.9	[3,7]
$D_{i}$	The symptomatic infectious period	2.9	2.9	2.9	2.9	2.9	[7,14]
g	The average waiting time for quarantined patients (day)	3.5	1.7	2	1.7	1.2	Fitted
$D_{h}$	The average hospitalization period	10	10	10	10	10	[4]
N	The total population in Wuhan	10,000,000	10,000,000	10,000,000	10,000,000	10,000,000	[7,40]
n	The daily domestic inbound and outbound travelers in Wuhan	500,000	800,000	0	0	0	[7,40]

The order of parameters was sorted according to Equation (1).

**Table 2 ijerph-18-11753-t002:** Parameters of estimating the first three parts of economic losses.

Meaning		Value	Reference
$Loss_{treatment}$ (treatment costs)			
$I_{i}$	The cumulative number of infected patients under various control measures (*r* and *g*)		Simulated
$P_{1}$	The probability of a COVID-19 infected patient being admitted to the Intensive Care Unit (ICU)	26.1(%)	[4]
$P_{2}$	The probability of Acute Respiratory Distress Syndrome (ARDS) after a COVID-19 infected patient is admitted to the Intensive Care Unit (ICU)	61.1(%)	[4]
$C_{1}$	The average treatment costs of a COVID-19 infected patient with no complications	67,360 (yuan)	[44]
$C_{2}$	The average treatment costs of a COVID-19 infected patient with complications	94,986 (yuan)	[44]
$C_{3}$	The average treatment cost of a COVID-19 infected patient with severe complications	140,006 (yuan)	[44]
$Loss_{quarantine}$ (Quarantine-related expenditures)			
$Q_{basic}$	The cumulative number of quarantined patients during the COVID-19 outbreak in Wuhan	26,893 (people)	Equation (1)
S	The total expenditures for disaster prevention and emergency management in Hubei province in the first quarter of 2020	15.68 (billion yuan)	[45]
$Q_{i}$	The cumulative number of quarantined patients under various control measures (*r* and *g*)		Simulated
$Loss_{inoperable}$ (Inoperable losses)			
$T_{I}$	The average time that infected patients received treatment and self-isolation	8 + 10 + 14 = 32(days)	[4]
$T_{Q}$	The average time that quarantined patients received quarantine and self-isolation	14 + 14 = 28(days)	Assumed
$GDP_{per}$	The per capita daily GDP in 2019	198(yuan)	[46]

**Table 3 ijerph-18-11753-t003:** Parameters of estimating social distancing losses.

	Meaning	Value	Reference
$r_{0}$	The effective number of daily contacts before the COVID-19 outbreak	13.96 (people)	Equation (5)
$C_{i}$	The contact’s number matrix with intervals of five years of age in China		[47]
$P_{i}$	The proportion of the age structure with intervals of five years of age in China in 2019		[46]
$\bar{r}$	The weighted value of the effective number of daily contacts during the COVID-19 outbreak	4.43 (people)	Equation (6)
$r_{i}$	The effective number of daily contacts in each stage of the COVID-19 epidemic in Wuhan		Estimated in Table 1
$D_{i}$	The proportion of the duration in each stage of the COVID-19 epidemic in Wuhan		[7]
$\frac{\bar{r}}{r_{0}}$	The reduction in contact activity caused by the COVID-19 outbreak in Wuhan	31.73 (%)	Equations (5) and (6)
α	The degree of nonlinearity of each industry sector’s response to the reduction in contact activity		Equation (8)
$C_{2019}$	The cumulative output value of each industry sector in the Hubei province in the first quarter of 2020		[45]
$GDP_{p}$	The actual growth rate of the industry sector in the Hubei province in the first quarter of 2020		[45]
$\tilde{GDP_{p}}$	The expected growth rate of the industry sector in the Hubei province in the first quarter of 2020		Estimated with the ARIMA model [48]
$Loss_{distancing}$ (Social distancing losses)			
$GDP_{p}{}^{r}$	The expected growth rate of each industry sector under the different level of contact activity		Equation (9)
r	The effective number of daily contacts	$(0,\;r_{0}$)	Assumed
$\frac{r}{r_{0}}$	The different level of contact activity	(0, 1)	Assumed
P	The percentage of the output value in 2019 Wuhan to the output value in the 2019 Hubei province	35.37 (%)	[45]

## Data Availability

The data used in this paper were collected from The Chinese Centers for Disease Control and Prevention, Hubei Provincial Statistics Bureau. The data are available on the respective websites: http://2019ncov.chinacdc.cn/2019-nCoV/ (accessed on 16 September 2021). http://tjj.hubei.gov.cn/tjsj/sjkscx/tjyb/tjyb2021/ (accessed on 16 September 2021).

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
