# Peer review of "Estimating Economic Losses Caused by COVID-19 under Multiple Control Measure Scenarios with a Coupled Infectious Disease—Economic Model: A Case Study in Wuhan, China"

_ijerph, 2021, doi:10.3390/ijerph182211753_

Round 1

Reviewer 1 Report

The authors propose a modeling framework to estimate the economic losss caused by COVID-19 in Wuhan, China.Their study addresses an important and hot practical issue. The results have good reference value. I have the following concerns for the editors and authors to consider.

1: The value of parameter n in model (1) should be double checked. It seems that the corresponding expression could not represent daily inbound/outbound population size.

2: There are too many paramters that need to be estimated. Sensitivity and uncertain analysis should be given for the output.

3: The fitting technique is not introduced in the paper.

4: It could be better to unify the  notation. "COVID-19" vs "SARS-CoV-2".

Author Response

Thanks for you comments. Please see the attachment.

Reviewer 2 Report

The article “Estimating Economic Losses Caused by COVID-19 Under Multiple Control Measure Scenarios with A Coupled Infectious Disease – Economic Model: a case study in Wuhan, China” is well-written. Its methodology is adequately demonstrated, and the results are well-informed by robust estimation models.

There are some areas that can be improved to strengthen the paper’s arguments.

Literature Review: The article needs an appropriate Literature Review section, especially for the Susceptible-Exposed-Infected-Recovered infectious disease model and the Economic loss model. Moreover, similar studies regarding estimating economic loss in a catastrophic event should also be presented.

There are several assumptions that need supporting citations. For instance, from lines 202-206:

“Among the above four parts of the economic losses, the first three can be assumed to have a linear relationship with the number of patients, but the last one, social distancing losses, are linear to neither the number of patients nor the number of daily contacts. Consequently, we assume a non-linear relationship between the number of daily contacts and social distancing losses.”

The Discussion needs to use some comparison with previous findings, as well as situations from other countries:

https://www.mdpi.com/2071-1050/12/7/2931

https://set.odi.org/wp-content/uploads/2020/05/Can-the-digital-economy-help-mitigate-the-economic-losses-from-COVID-19-in-Kenya.pdf

https://jamanetwork.com/journals/jama/article-abstract/2771764

Finally, it would be more convincing if the paper considers the contexts where patients might be subjected to different socio-economic conditions because these will affect the sustainability of the public health system as well as the perceptions of the welfare system for the populace. Please have a look at an example:

On sensitivity to healthcare costs: https://www.nature.com/articles/s41599-018-0127-3

All in all, the paper is meaningful and worth considering. And I trust that its value will be further enhanced once the suggested revisions are taken seriously. Then it will be a valuable piece the literature.

Best wishes

Author Response

(The authors gave the same response as above.)

Round 2

Reviewer 2 Report

Dear Authors,

Thank you very much for your revised submission. Having examined the paper again, I trust that it is now in good shape and ready for publication.

Congratulations on the work well done.

Best wishes